# Bayesian Learning of Probabilistic Dipole Inversion for Quantitative Susceptibility Mapping

**Jinwei Zhang**[1,2]                                                  JZ853@CORNELL.EDU
[1] *Department of Biomedical Engineering, Cornell University, Ithaca, NY, USA*
[2] *Department of Radiology, Weill Medical College of Cornell University, New York, NY, USA*

**Hang Zhang**[2,3]                                                  HZ459@CORNELL.EDU
**Mert Sabuncu**[1,2,3]                                           MSABUNCU@CORNELL.EDU
[3] *Department of Electrical and Computer Engineering, Cornell University, Ithaca, NY, USA*

**Pascal Spincemaille**[2]                                    PAS2018@MED.CORNELL.EDU
**Thanh Nguyen**[2]                                          TDN2001@MED.CORNELL.EDU
**Yi Wang**[1,2]                                                YIWANG@MED.CORNELL.EDU

## Abstract

A learning-based posterior distribution estimation method, Probabilistic Dipole Inversion (PDI), is proposed to solve quantitative susceptibility mapping (QSM) inverse problem in MRI with uncertainty estimation. A deep convolutional neural network (CNN) is used to represent the multivariate Gaussian distribution as the approximated posterior distribution of susceptibility given the input measured field. In PDI, such CNN is firstly trained on healthy subjects' data with labels by maximizing the posterior Gaussian distribution loss function as used in Bayesian deep learning. When tested on new dataset without any label, PDI updates the pre-trained CNN's weights in an unsupervised fashion by minimizing the *KullbackLeibler* divergence between the approximated posterior distribution represented by CNN and the true posterior distribution given the likelihood distribution from known physical model and pre-defined prior distribution. Based on our experiments, PDI provides additional uncertainty estimation compared to the conventional MAP approach, meanwhile addressing the potential discrepancy issue of CNN when test data deviates from training dataset.

**Keywords:** Bayesian deep learning, variational inference, convolutional neural network, quantitative susceptibility mapping

## 1. Introduction

Consider the following biomedical imaging model:

$$y = Ax + n \tag{1}$$

where $A$ the forward imaging system model, $x$ the underlining biomedical image variable, $n$ the system noise, and $y$ the measured data/signal variable. Because of the intrinsic ill-posedness of forward imaging operator $A$, prior term is needed in the following Maximum a posteriori (MAP) estimation problem (Kaipio and Somersalo, 2006):

$$\hat{x} = \arg\max_x p(x|y) \propto p(y|x)p(x) \tag{2}$$

where $p(x)$ is the prior term to regularize the inverse problem. Assuming zero mean Gaussian noise with covariance matrix $\Sigma$, Eq. 2 is equivalent to the following minimum $-\log p(x|y)$ problem:

$$\hat{x} = \arg\min_x ||Ax - y||^2_{\Sigma^{-1/2}} + R(x) \tag{3}$$

where $R(x) = -\log p(x)$. Convex optimization solvers have been widely used to solve Eq. 3 with both accuracy and efficiency, such as quasi-newton method (Dennis and Moré, 1977), alternating direction method of multipliers (ADMM) (Boyd et al., 2011) and primal dual method (PD) (Chambolle and Pock, 2011).

In recent years, posterior distribution estimation in imaging inverse problems has been a new topic in medical imaging field (Repetti et al., 2019; Chappell et al., 2009; Tezcan et al., 2018), in which random variable's variance is provided from posterior distribution to measure the uncertainty of the solution. However, posterior distribution estimation requires complicated or even intractable integral from Bayes formula, therefore sampling or approximation method is used to reduce the computational cost and intractability of the problem. Markov chain Monte Carlo (MCMC) (Andrieu et al., 2003) and variational inference (VI) (Bishop, 2006) are two common frameworks in Bayesian estimation problem. In MCMC, efficient sampling methods are used to get random samples from posterior distribution. After proper sampling procedure, random samples can represent an empirical distribution which resembles the true distribution. However, in imaging inverse problem, the computational cost of approximating integrals for Bayesian estimation is often several magnitude higher than the optimization method of MAP estimation, suffering from curse of dimensionality (Pereyra, 2017).

An alternative approach is to use VI, in which an approximation distribution is proposed with specific function form and unknown parameters, and then optimization algorithm is implemented (for example, expectation-maximization (EM) algorithm (Blei et al., 2017)) to learn these parameters by minimizing the divergence between true posterior and approximate posterior. After learning/fitting, the approximate posterior represents the true posterior. However, approximation quality is determined by the flexibility of approximate function form and trainable parameters. More complicated approximate function has better representation ability, however, the computational cost becomes higher. Therefore, the trade-off between number of trainable parameters of approximate function and learning/fitting efficiency needs careful consideration for VI.

Over the past years, thanks to the advances of deep learning, using deep neural network as the approximate function has become a new trend in VI, especially for generative models (Rezende et al., 2014; Kingma and Welling, 2013), in which low dimensional latent space variables are modeled and encoder and decoder networks are built to approximate the latent variable distribution conditioned on observed data and reversely observed data distribution conditioned on latent variable. Due to the approximation and generalization

power of deep neural network with millions of trainable weights, neural network can approximate any function/distribution with high accuracy. In addition, advanced stochastic optimization algorithms such as ADAM (Kingma and Ba, 2014) have been proposed for efficient backpropagation in network weights' updating. Another topic related to the posterior distribution estimation with deep learning is discussed in Bayesian deep learning framework (Kendall and Gal, 2017), in which data uncertainties are captured by maximizing the posterior distribution with the labels as the samples assuming they follow multivariate Gaussian.

In this paper, we come up with a framework by combining Bayesian deep learning to model data uncertainties and VI with deep learning to approximate true posterior distribution, and apply it to one important imaging inverse problem in MRI: *quantitative susceptibility mapping* (QSM) (de Rochefort et al., 2010; Wang and Liu, 2015), which has the advantages of mapping iron decomposition (Wang et al., 2017) and calcification (Chen et al., 2014). Assuming multivariate Gaussian represented by a CNN as the posterior distribution of susceptibility given the input local field, golden standard susceptibility maps COSMOS (Calculation Of Susceptibility through Multiple Orientation Sampling (Liu et al., 2009)) are used to train such CNN with a maximal posterior loss function. With physical model-based likelihood term and delicately designed prior term, the pre-trained CNN can be enhanced when tested on patient dataset by minimizing the Kullback-Leibler (KL) divergence between true posterior distribution and approximation distribution represented by such CNN. Our experimental results show the proposed method gives mean and variance estimation of the solution automatically, and yields optimal results compared to two types of benchmark methods: deep learning QSM ((Yoon et al., 2018; Zhang et al., 2020)) and *maximum a posteriori* (MAP) QSM with convex optimization (Liu et al., 2012; Kee et al., 2017; Milovic et al., 2018), both of which do not provide uncertainty estimation.

## 2. Method

### 2.1. Modeling

In Magnetic Resonance Imaging (MRI), the forward model of generating relative local field $b$ from tissue susceptibility $\chi$ is:

$$b = d * \chi + n \tag{4}$$

where $*$ denotes convolution operation, $d$ denotes *dipole kernel*, which is ill-posed inherent in the structure of dipole convolution operator. $b$ is derived from multi-echo gradient echo MR signal with noise $n$ estimated as well. The inverse problem of estimating $\chi$ from measured $b$ is called *Quantitative Susceptibility Mapping* (QSM). From convolution theory, the forward convolution process in Eq. 4 is equivalent to the following Fourier space multiplication process:

$$b = F^H D F \chi + n \tag{5}$$

where $F$ is Fourier matrix, $D$ is dipole kernel in Fourier space. Eq. 5 is computationally friendly since Fast Fourier Transform (FFT) can be used efficiently. We will use forward model in Eq. 5 for computation in this paper.

One successful approach to solving QSM from single orientation field $b$ is MEDI (Morphology enabled dipole inversion) (Liu et al., 2011b, 2012), where weighted total variation regularization was imposed onto the area except tissue in brain and the following MAP estimation is deployed:

$$\hat{\chi} = \arg\min_{\chi} ||W(F^H DF\chi - b)||_2^2 + \lambda||M\nabla\chi||_1 \tag{6}$$

where $W$ is derived from observation noise covariance matrix and $M$ is gradient's weight to penalize only region's outside brain tissues. Computational methods for solving Eq. 6 is reviewed in (Kee et al., 2017).

Starting from forward model in Eq. 5, we develop the fully probabilistic model of QSM and use approximate Bayesian inference to solve this problem. We assume conditional distribution of field $b$ given susceptibility $\chi$ as a Gaussian distribution:

$$p(b|\chi) = \mathcal{N}(b|F^H DF\chi, \Sigma_{b|\chi}) \tag{7}$$

where we assume $n \sim \mathcal{N}(0, \Sigma_{b|\chi})$ with $\Sigma_{b|\chi}$ diagonal in Eq. 5. The prior distribution from Eq. 6 reads:

$$p(\chi) \propto e^{-\lambda||M\nabla\chi||_1}. \tag{8}$$

Other types of prior distributions can also be applied. Because of the intractability of estimating the posterior distribution $p(\chi|b) = p(b|\chi)p(\chi)/\int_{\chi} p(b|\chi)p(\chi)d\chi$ in most cases, approximate posterior distribution $q(\chi|b) = \mathcal{N}(\mu_{\chi|b}, \Sigma_{\chi|b})$ with diagonal covariance matrix is assumed to approximate the true posterior distribution $p(\chi|b)$. In this work, we use a dual-decoder network architecture (Figure 1) extended from 3D U-Net (Ronneberger et al., 2015; Çiçek et al., 2016) to represent the approximate posterior $q_\psi(\chi|b)$, with each decoder's output representing mean $\mu_{\chi|b}$ and variance $\Sigma_{\chi|b}$ map, respectively.

## 2.2. Supervised Bayesian Training

For training dataset with COSMOS as golden standard labels, we can treat these labels as samples from the true posterior distribution, and train the approximate distribution $q_\psi(\chi|b)$ in a supervised fashion with the following MAP loss function:

$$-\log q_\psi(\chi_i|b_i) = \frac{1}{2}(\chi_i - \mu_{\chi|b_i})^T \Sigma_{\chi|b_i}^{-1}(\chi_i - \mu_{\chi|b_i}) + \frac{1}{2}\ln|\Sigma_{\chi|b_i}|, \tag{9}$$

where $\{b_i, \chi_i\}$ denote the i-th input and label data pair in the training dataset. Note that because of multiple orientations' scanning for COSMOS, this dataset is quite limited and usually only on healthy subjects. We denote this supervised Bayesian learning approach as Probabilistic Dipole Inversion (PDI).

## 2.3. Unsupervised Variational Inference

After training using COSMOS data with loss function Eq. 9 and obtaining optimal parameters $\psi^*$, given a test local field $b'$, we can simply estimate $p(\chi|b')$ as $q_{\psi^*}(\chi|b')$. However, for

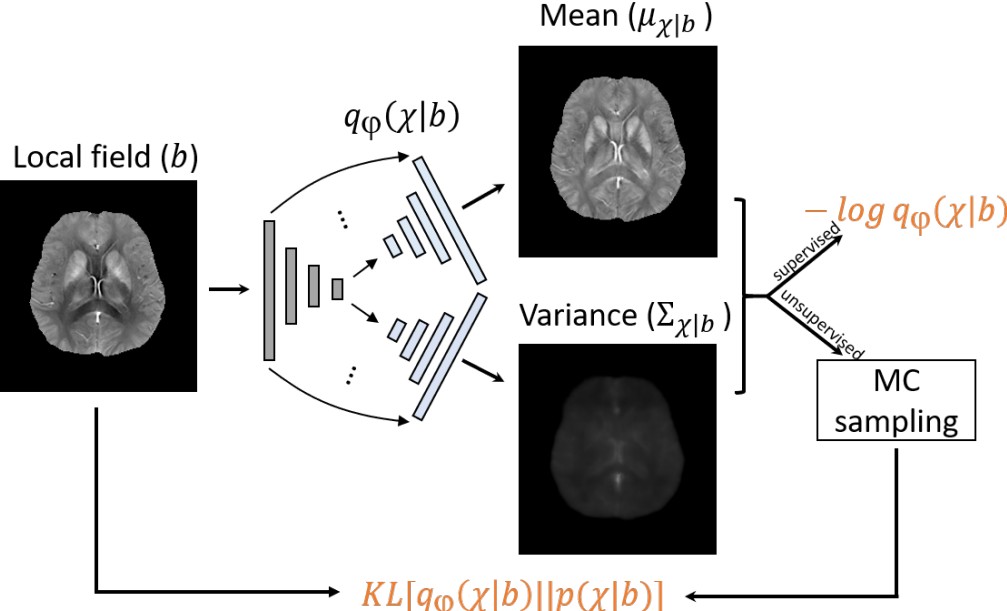

Figure 1: Network architecture of the proposed method. Dual decoders' outputs represent mean and variance maps. COSMOS dataset was used to do supervised Bayesian training via MAP in Eq. 9. Unsupervised VI with MC sampling in Eq. 11 and 12 was applied on other test dataset.

new test dataset which has input field $b'$ deviating from COSMOS training dataset (such as having new pathologies), inferior outputs could be produced. In this case, $q_{\psi^*}(\chi|b')$ can be enhanced by deploying variational inference on a subset of this new test dataset as another training set. specifically, the pre-trained approximation $q_\psi(\chi|b')$ with weights $\psi$ initialized as $\psi^*$ can be fine-tuned by minimizing the KL divergence between $p(\chi|b')$ and $q_\psi(\chi|b')$:

$$
\begin{aligned}
& \mathrm{KL}[q_\psi(\chi|b')||p(\chi|b')] \\
&= \mathbb{E}_q[\log q_\psi(\chi|b') - \log p(\chi|b')] \\
&= \mathbb{E}_q[\log q_\psi(\chi|b') - \log p(\chi, b')] + \log p(b') \\
&= \mathrm{KL}[q_\psi(\chi|b')||p(\chi)] - \mathbb{E}_q[\log p(b'|\chi)]
\end{aligned}
\tag{10}
$$

where the first term in the last equation above imposes the posterior to be similar to the prior, and the second term encourages data consistency in QSM foward model. Applying the prior term defined in Eq. 8 and likelihood term in Eq. 7, KL divergence in Eq. 10 becomes:

$$
\begin{aligned}
& \mathrm{KL}[q_\psi(\chi|b')||p(\chi|b')] \\
&= -\frac{1}{2}\ln|\Sigma_{\chi|b'}| + \frac{1}{2K}\sum_{k=1}^{K}\lambda\|M\nabla\chi_k\|_1 + \frac{1}{2K}\sum_{k=1}^{K}(\chi_k * d - b')^T\Sigma_{b'|\chi}^{-1}(\chi_k * d - b')
\end{aligned}
\tag{11}
$$

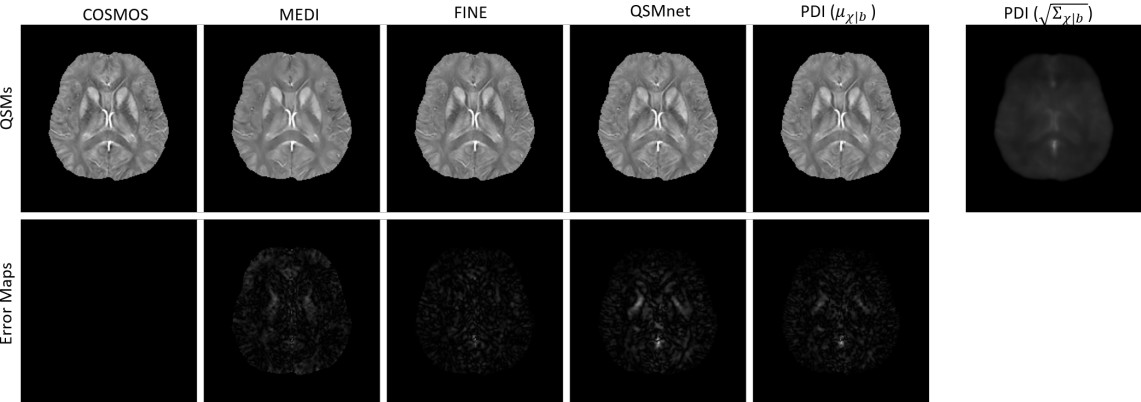

Figure 2: Reconstructions (first row) and error maps (second row) of one COSMOS test subject in one orientation, with COSMOS as the golden standard. FINE gives the best reconstruction at the expense of significantly increased computational time. The other three methods have comparable results. The standard deviation map (last column) provided by PDI resembled its error map, with high uncertainties/errors locating at sagittal sinus and globus pallidus.

where $-\mathbb{E}_q[\ln p(\chi)]$ in $\mathrm{KL}[q_\psi(\chi|b')||p(\chi)]$ and $-\mathbb{E}_q[\log p(b'|\chi)]$ are calculated through Monle Carlo (MC) sampling with $\chi_k$ sampled from $q_\psi(\chi|b')$ because of the intractability of both expectations. We denote the fine-tuned approximate distribution with Eq. 11 as PDI-VI1.

Another possible prior term for $\chi$ is simply constant prior $p(\chi) \propto c$, which means no prior information is given regarding the distribution of $\chi$. With such non-informative prior, the corresponding loss function is simply:

$$\mathrm{KL}[q_\psi(\chi|b')||p(\chi|b')] = -\frac{1}{2}\ln|\Sigma_{\chi|b'}| + \frac{1}{2K}\sum_{k=1}^{K}(\chi_k * d - b')^T \Sigma_{b'|\chi}^{-1}(\chi_k * d - b') \quad (12)$$

We denote the fine-tuned approximate distribution with Eq. 12 as PDI-VI2.

## 3. Experiments

MRI was performed on 7 healthy subjects with 5 brain orientations using a 3T GE scanner equipped with a multi-echo 3D gradient echo (GRE) sequence. Acquisition matrix was $256 \times 256 \times 48$ and voxel size was $1 \times 1 \times 3$ mm$^3$. The input local tissue field data $b$ was generated by sequentially deploying non-linear fitting across multi-echo phase data (Kressler et al., 2009), graph-cut based phase unwrapping (Dong et al., 2014) and background field removal (Liu et al., 2011a). COSMOS reconstruction (Liu et al., 2011b) was calculated from 5 orientations' GRE imaging and was used as the gold standard label in the experiment. A second dataset was obtained by performing single orientation GRE MRI on 8 patients with intracerebral hemorrhage (ICH), which were acquired using the same scanner and imaging parameters as above.

Network architecture is shown in Figure 1. Dual decoders' outputs were used to represent mean and variance maps in the posterior susceptibility distribution given input local field. The 3D convolutional kernel size was $3 \times 3 \times 3$. The numbers of filters from the highest

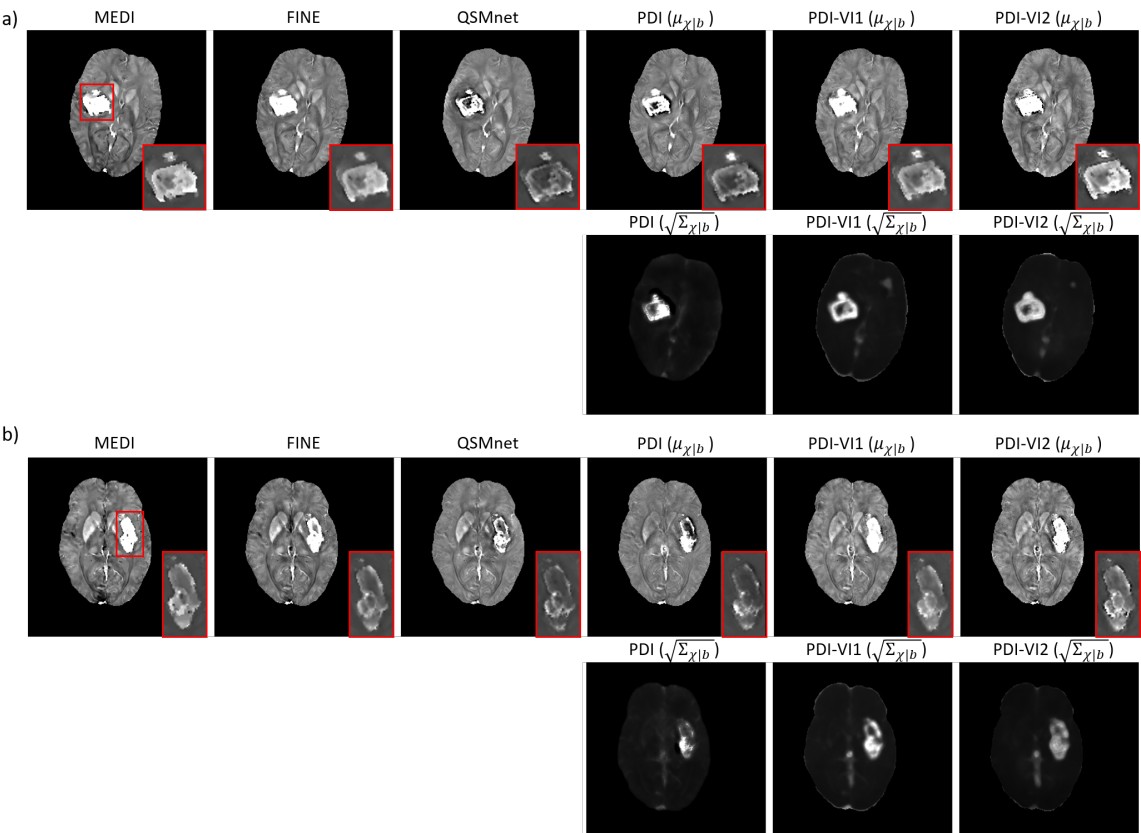

Figure 3: Reconstructions (first row in (a) and (b)) and standard deviation maps (second row in (a) and (b)) of two ICH patients. Compared to MEDI and FINE, underestimation issue inside hemorrhage happened on QSMnet and PDI. This issue was reduced in PDI-VI1 and PDI-VI2 by fine-tuning the pre-trained network using unsupervised variational inference. High variance inside the hemorrhage was consistent with high measured noise in the same region.

feature level to the lowest were 32, 64, 128, 256 and 512, respectively. Batch normalization (Ioffe and Szegedy, 2015), max pooling for downsampling and deconvolution operation for upsampling were used. For COSMOS dataset, 4/1 subjects (20/5 brain volumes) were used as training/validation dataset, with augmentation by in-plane rotation of $\pm 15°$. Each brain volume data in the training and validation dataset was divided into 3D patches with patch size $64 \times 64 \times 32$ and extraction step $21 \times 21 \times 11$. The remaining 2 subjects (10 brain volumes in total) were used for testing. For ICH patients dataset, 5/1 subjects were used as training/validation dataset for PDI-VI1 and PDI-VI, and the remaining 2 subjects were used for testing.

Loss function in Eq. 9 was applied for supervised Bayesian training on COSMOS dataset with ADAM optimizer (Kingma and Ba, 2014) (learning rate: $10^{-3}$, Number of epochs: 60), yielding trained network $q_{\psi^*}(\chi|b)$. The outputs of $q_{\psi^*}(\chi|b)$ were denoted as PDI. Initialized with the pre-trained PDI using COSMOS training data, unsupervised variational inference with loss function Eq. 11 and 12 was also applied on ICH dataset using ADAM optimizer (learning rate: $10^{-3}$, Number of iterations: 100). MC sampling size $K$ was chosen as 5 due

|  | pSNR | RMSE | SSIM | HFEN | GPU time (s) |
|---|---|---|---|---|---|
| MEDI (Liu et al., 2012) | 46.39 | 41.16 | 0.9569 | 31.30 | 17.54 |
| FINE (Zhang et al., 2020) | 48.12 | 33.66 | 0.9789 | 31.97 | 65.42 |
| QSMnet (Yoon et al., 2018) | 46.35 | 41.29 | 0.9705 | 43.31 | 0.60 |
| PDI (Eq. 9) | 47.77 | 35.08 | 0.9772 | 35.17 | 0.61 |

Table 1: Mean quantitative metrics of 10 test COSMOS brains reconstructed by different methods. FINE gives the best reconstruction at the expense of significantly increased computational time. The other three methods have comparable results.

to limited GPU memory and reparameterization trick (Kingma and Welling, 2013) was used for MC sampling in order to do backpropagation. The outputs were denoted as PDI-VI1 and PDI-VI2, respectively. The whole brain volume was fed into the network during testing, including unsupervised variational inference step. We implemented the proposed method using PyTorch (Python 3.6) on an RTX 2080Ti GPU.

For COSMOS test dataset, we compared PDI with MAP estimation MEDI (Liu et al., 2012) and two deep learning reconstructions QSMnet (Yoon et al., 2018) and FINE (Zhang et al., 2020). Reconstruction maps of one orientation from one test subject are shown in Figure 2 ([-0.15ppm, 0.15ppm]). Quantitative metrics of each reconstruction method averaged among 10 test brains are shown in Table 1. FINE gave the best overall quantitative results; However, it overfitted to every test case by minimizing the fidelity loss, which had the major drawback of significantly increased computational time. PDI gave slightly better results than MEDI and QSMnet, meanwhile achieved fast inference time on GPU comparable to QSMnet. In figure 2, error map of PDI's mean output $\mu_{\chi|b}$ was coincident with PDI's standard deviation output $\sqrt{\Sigma_{\chi|b}}$, with high uncertainty/error happening at sagittal sinus and globus pallidus.

For ICH test dataset, PDI-VI1 and PDI-VI2 were also performed and compared. Two representative ICH patients' QSMs are shown in Figure 3 ([-0.6ppm, 1.5ppm] for zoomed-in hemorrhage). Compared to MEDI and FINE which had hyperintensity inside the hemorrhage, both QSMnet and PDI suffered from underestimation issue inside this region, which might result from the fact that such pathology was not encountered during training since long scan COSMOS was not practical for the patients. After PDI-VI1 and PDI-VI2, such underestimation issue was reduced and variance maps' structures inside the hemorrhage were also better depicted. High uncertainties inside hemorrhage as shown in Figure 3 were consistent with high local field noise level which was approximately proportional to the underlining susceptibility values.

## 4. Conclusion

We developed a Bayesian dipole inversion framework for quantitative susceptibility mapping by combining variational inference and Bayesian deep learning. Our method generated high fidelity susceptibility maps meanwhile provided uncertainty quantifications. When applied to other datasets not encountered during training, the proposed method was able to correct the undesirable outputs in an unsupervised fashion based on variantional inference principle.

## Acknowledgments

This research was supported in part by National Institute of Health (R01NS090464, R01NS095562, R01NS105144, R01DK116126, R01CA181566, S10OD021782, 1R21AG050122, R01LM012719 and R01AG053949), National Science Foundation (1748377 and 1707312) and National Multiple Sclerosis Society (RR-1602-07671).

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
