# OpenReview forum: "Bayesian Learning of Probabilistic Dipole Inversion for Quantitative Susceptibility Mapping"
_MIDL.io/2020/Conference — MIDL 2020_

### Official Review · AnonReviewer2 · 2020-03-13
**A very good contribution to MIDL**

**Rating:** 4
**Confidence:** 4
**Recommendation:** Oral

**Summary:**

A Bayesian approach to solving quantitative susceptibility mapping (QSM) inverse problem in MRI is proposed. The authors propose to approximate the posterior distribution of tissue susceptibility using a diagonal-covariance Gaussian with mean, variance predicted by a neural network.

The overall framework is reminiscent of VAEs, except with a known generative model given by the physics of the problem. From that analogy the inverse problem is approached as that of learning an optimal encoder. The approximating network is pretrained in a supervised manner on healthy subject data with known susceptibility, local field pairs; and fine-tuned on test subjects using KL-divergence minimization.

The experimental validation is still preliminary, conducted on the order of 20(?) subjects.

**Strengths:**

The paper is very well written, the introduction and description of the method are of high quality. The approach is sound. The experimental validation seems limited but the results that are shown are again, well presented and interesting.

I have limited knowledge on the application itself and cannot fully judge, but the paper provides the necessary material to understand the task and challenges at a high level.

Overall the paper is very pleasant to read and the content of the paper / validation is well aligned with the original claims.

**Weaknesses:**

I do not really have important weaknesses to point out. I have a few minor questions that come to mind about choices made in the paper, but they are not essential to address in the rebuttal (see comments).



**Detailed Comments:**

I would be curious about additional insight into the "two stage" supervised/unsupervised procedure. What guides this choice? Is there very limited data available? One may think of fine-tuning the network parameters using all cases with unknown local field together (rather than fine-tuned weights per test case). Does it perform worse? Same question when training jointly on all available data, with known or unknown field (using Eq. 9 or 11 as appropriate).

Is the architecture of the decoders (following the paper's terminology) limiting? In terms of formulation, the approach boils down to an approximate inverse problem (approximate solution to Eq. 6 or Eq. 7-8). It would be interesting to know if there are significant gains, in run time (even when using VI for fine-tuning) and/or beyond run time gains.

Is there a trade-off in the number of iterations for which single-test case fine-tuning is performed?

Finally, it would be interesting to extend the approach with a partly learnable prior, so that the approach can fully leverage training data.

**Justification Of Rating:**

The paper is very well written, the introduction and description of the method are of high quality. The approach is sound. The experimental validation seems limited but the results that are shown are again, well presented and interesting.

I have limited knowledge on the application itself and cannot fully judge, but the paper provides the necessary material to understand the task and challenges at a high level.

Overall the paper is very pleasant to read and the content of the paper / validation is well aligned with the original claims.

**Paper Type:**

methodological development

**Special Issue:**

yes

---

> ### Author Response · Authors · 2020-03-27
> **Modifying the unsupervised VI step by fine-tuning the network parameters on a hemorrhage dataset**
>
>
> We thank the reviewer for the appreciation of our work. The corresponding responses are listed below:
>
> 1. I would be curious about additional insight into the "two stage" supervised/unsupervised procedure. What guides this choice? Is there very limited data available? One may think of fine-tuning the network parameters using all cases with unknown local field together (rather than fine-tuned weights per test case). Does it perform worse? Same question when training jointly on all available data, with known or unknown field (using Eq. 9 or 11 as appropriate).
>
> Reply: Since the hemorrhage dataset on which we deployed VI with Eq. 11 or 12 was very limited (with only 5 cases by the time we submitted the manuscript), a natural way of applying VI we first came up with was to do it one by one, i.e., training and test on the same data, and the computational cost of doing so for all 5 cases was manageable.
> Thanks to the reviewer’s suggestion and 3 more acquired hemorrhage patient data, we will make the change to deploy the VI step using Eq. 11 or 12 to train our model on 5 cases, validation on 1 case and test on independent 2 cases, shown in Fig. 3. Comparable results are achieved now and these new results can be found in our updated Fig. 3: http://gdurl.com/YTFy (We also add new benchmark FINE (Zhang et al., 2020) for comparison as requested by anonReviewer4).
> Future work will include exploring a semi-supervised learning strategy that uses all available data together with Eq. 9 or Eq. 11/12 as loss functions.
>
>
> 2. Is the architecture of the decoders (following the paper's terminology) limiting? In terms of formulation, the approach boils down to an approximate inverse problem (approximate solution to Eq. 6 or Eq. 7-8). It would be interesting to know if there are significant gains, in run time (even when using VI for fine-tuning) and/or beyond run time gains.
>
> Reply: In our experience, the adopted U-Net decoder architecture performed very well for image-to-image tasks. Other deep learning QSM methods (Yoon et al., 2018; Zhang et al., 2020) also use a similar architecture. The run time of forward pass in this architecture was quite fast (less than one second, see new Table 1: http://gdurl.com/iR-c ). However, the run time of VI fine-tuning was slow (~ 5 mins for each hemorrhage case). Uncertainty estimation serves as another gain.
>
>
> 3. Is there a trade-off in the number of iterations for which single-test case fine-tuning is performed?
>
> Reply: In our experiments, at least 100 iterations were needed to ‘correct’ the maps. But too many iterations might impair the maps.
>
>
> 4.Finally, it would be interesting to extend the approach with a partly learnable prior, so that the approach can fully leverage training data.
>
> Reply: Thank you for this valuable advice. Possible partly learnable prior could be some additional density estimation network trained by variational autoencoder or adversarial autoencoder on COSMOS data, and then use ELBO($\chi$) evaluated by such density network to approximate the prior distribution $ \log p(\chi)$. We will explore this in the future work.

---

### Official Review · AnonReviewer4 · 2020-03-14
**An interesting application of Bayesian machine learning, while some details are unclear.**

**Rating:** 3
**Confidence:** 3
**Recommendation:** Poster

**Summary:**

This paper proposes a supervised Bayesian learning approach, namely Probabilistic Dipole Inversion (PDI), to model data uncertainties for the quantitative susceptibility mapping (QSM) inverse problem in MRI. This paper employs the dual-decoder network architecture to represent the approximated posterior distribution and uses the MAP loss function to train the approximate distribution when the labels exist. When having new pathologies in test data, the proposed method minimizes the KL divergence between the approximated posterior distribution and the true posterior distribution based on the variational inference principle to correct the outputs. Experiments show that the proposed method can capture uncertainty compared to other methods.

**Strengths:**

1. This paper proposes a supervised Bayesian learning approach, namely Probabilistic Dipole Inversion (PDI), to model data uncertainties for the quantitative susceptibility mapping (QSM) inverse problem in MRI. The motivation is good.
2. Experiments show that the proposed method can capture uncertainty compared to other methods.

**Weaknesses:**

1.	The authors point that they use the forward model in Eq.5 for computation in this paper. However, PDI-VI1 and PDI-VI2 represented by Eq. 11 and Eq. 12, respectively, use the forward model in Eq.4. The authors should unify the expression form according to the actual situation.
2.	For the unsupervised variational inference case, it is not clear whether the Fourier matrix $F$, the dipole kernel $D$, and the noise covariance matrix $\Sigma_{b|\chi}$ in the likelihood term are parameters that need to be optimized or have been determined before training.
3.	This paper uses the dual-decoder network architecture to represent the approximated posterior distribution. It’s better to provide the specific network architecture adopted in the experiment，rather than just a simple schematic.
4.	This paper uses three quantitative metrics, namely RMSE, SSIM, and HFEN, to measure the reconstruction quality. Please give the full names of the three quantitative metrics. Other papers (Yoon et al., 2018; Zhang et al., 2020) also use the peak signal-to-noise ratio (pSNR) to measure the reconstruction quality. The performance on the pSNR is better to show in the experimental results. And please further explain why the performance of MEDI is better than PDI and QSMnet on the HFEN metric.
5.	This paper states that the experimental results show the proposed method yields optimal results compared to two types of benchmark methods: deep learning QSM (Yoon et al., 2018; Zhang et al., 2020) and maximum a posteriori (MAP) QSM with convex optimization (Liu et al., 2012; Kee et al., 2017; Milovic et al., 2018). And they compare PDI with MEDI (Liu et al., 2012) and QSMnet (Yoon et al., 2018). It is better to add experiments that compare with other more advanced benchmark methods, such as FINE (Zhang et al., 2020).
6.	Figure 3 shows the reconstructions and standard deviation maps of two ICH patients. please explain what the red rectangle highlights for a better understanding.
7.	The presentation should be improved. For example,  “In this paper, we come up with a framework by combining Bayesian deep learning to model data uncertainties and VI with deep learning to approximate true posterior distribution” , and “we developed a Bayesian dipole inversion framework for quantitative susceptibility mapping by combining variational inference and Bayesian deep learning. “, VI is just an inference method, which could be included in Bayesian ML or Bayesian DL.


**Detailed Comments:**

See above

**Justification Of Rating:**

This paper proposes a supervised Bayesian learning approach, namely Probabilistic Dipole Inversion (PDI), to model data uncertainties for the quantitative susceptibility mapping (QSM) inverse problem in MRI. The motivation is good and this is an interesting application of Bayesian learning.  The results look good, while some details are unclear. The presentation can be further improved.

**Paper Type:**

validation/application paper

**Questions To Address In The Rebuttal:**

1.	This paper uses three quantitative metrics, namely RMSE, SSIM, and HFEN, to measure the reconstruction quality. Please give the full names of the three quantitative metrics. Other papers (Yoon et al., 2018; Zhang et al., 2020) also use the peak signal-to-noise ratio (pSNR) to measure the reconstruction quality. The performance on the pSNR is better to show in the experimental results. And please further explain why the performance of MEDI is better than PDI and QSMnet on the HFEN metric.
2.	This paper states that the experimental results show the proposed method yields optimal results compared to two types of benchmark methods: deep learning QSM (Yoon et al., 2018; Zhang et al., 2020) and maximum a posteriori (MAP) QSM with convex optimization (Liu et al., 2012; Kee et al., 2017; Milovic et al., 2018). And they compare PDI with MEDI (Liu et al., 2012) and QSMnet (Yoon et al., 2018). It is better to add experiments that compare with other more advanced benchmark methods, such as FINE (Zhang et al., 2020).

**Special Issue:**

no

---

> ### Author Response · Authors · 2020-03-27
> **Adding pSNR metric and FINE benchmark**
>
>
> We thank the reviewer for pointing out the unclearness of this paper. The detailed responses are as follows:
>
> 1.This paper uses three quantitative metrics, namely RMSE, SSIM, and HFEN, to measure the reconstruction quality. Please give the full names of the three quantitative metrics. Other papers (Yoon et al., 2018; Zhang et al., 2020) also use the peak signal-to-noise ratio (pSNR) to measure the reconstruction quality. The performance on the pSNR is better to show in the experimental results. And please further explain why the performance of MEDI is better than PDI and QSMnet on the HFEN metric.
>
> Reply: We will add PSNR values for further comparison. Please check here for new results of Table 1: http://gdurl.com/iR-c . The reason why the baseline method FINE gives the best reconstruction results is that FINE overfits to every test case by minimizing the fidelity loss, which has the major drawback of significantly increased computational time.
> We will include more details on the definitions of the other metrics. HFEN (high-frequency error norm) is calculated as the L2 difference between Laplacian of a Gaussian (LoG) filtered reference and input volume, where LoG filter extracts the edges of the smoothed objects. In the MEDI formulation Eq. 6, gradient mask $M$ was obtained as the region outside tissue boundaries/edges, while gradient operator smoothed the region in $M$. This way, the brain was smoothed while the tissue edges were retained, which favored the HFEN metric defined above.
>
>
> 2. This paper states that the experimental results show the proposed method yields optimal results compared to two types of benchmark methods: deep learning QSM (Yoon et al., 2018; Zhang et al., 2020) and maximum a posteriori (MAP) QSM with convex optimization (Liu et al., 2012; Kee et al., 2017; Milovic et al., 2018). And they compare PDI with MEDI (Liu et al., 2012) and QSMnet (Yoon et al., 2018). It is better to add experiments that compare with other more advanced benchmark methods, such as FINE (Zhang et al., 2020).
>
> Reply: Thanks for the suggestion. We will add FINE results for comparison. Please check here for new results of Fig. 2: http://gdurl.com/zW03 .
> Another change we will make is to fine-tune the pre-trained network using Eq. 11 or 12 on a training set of hemorrhage cases (we now have an additional set of cases), instead of fine-tuning for every single hemorrhage patient case separately. Please see our response 3 to anonReviewer3 and response 1 to anonReviewer2 for details. The corresponding new results of Fig. 3 is: http://gdurl.com/YTFy .

---

### Official Review · AnonReviewer3 · 2020-03-19
**An adapted but not well validated method for solving quantitative susceptibility mapping**

**Rating:** 3
**Confidence:** 3
**Recommendation:** Poster

**Summary:**

This work introduces a Bayesian deep learning approach for solving Quantitative Susceptibility Mapping. Given a local field, the method generates a susceptibility map. Supervised and unsupervised learning are combined in order to generalise from healthy data to data with haemorrhage. The method is principled, fits with the underlying optimisation problem and achieves promising performance.

**Strengths:**

- the manuscript is well-written. I would like to congratulate the authors for their clarity.
- the proposed method is principled and relevant regarding the underlying optimisation problem.
- the solution seems easy to implement and efficient in practice.

**Weaknesses:**

Method
- The author argue that, given the "intrinsic ill-posedness" of the problem, "a prior term is needed". However, the term with the prior introduced in Eq 8 is removed in the final formulation (eq 12). This means that you assume that the network inherently induces some constraints that are beneficial for your problem. I don't see why it would be the case. Moreover, I suspect, as explained later, that better results are obtained without the regularisation because your model overfits.
- How do you estimate $\Sigma_{b'|\Chi)$ in eq. 12?

Experiments
-  it would seem that the unsupervised learning component via VI was trained and tested on the same data.
If so, this is for me a major weakness on this work.
The network may overfit on the testing data. Moreover, although you don't use any annotation for this task, this can be seen as an optimisation per subject. In this case, there is no clear advantage of using a deep learning approaches compared to other optimisation methods.

**Justification Of Rating:**

The paper is clear and the approach is principled.  The authors exploit Variationnal Inference with neural networks to solve the optimisation problem, which seems to be novel and a good idea given the formulation. However, I have concerns regarding the validation.

**Paper Type:**

methodological development

**Special Issue:**

no

---

> ### Author Response · Authors · 2020-03-27
> **Addressing the major weakness by fine-tuning on a hemorrhage dataset in the unsupervised VI step and then test**
>
>
> We thank the reviewer for the helpful advise. The detailed responses are as follows:
>
> 1. Method - The author argue that, given the "intrinsic ill-posedness" of the problem, "a prior term is needed". However, the term with the prior introduced in Eq 8 is removed in the final formulation (eq 12). This means that you assume that the network inherently induces some constraints that are beneficial for your problem. I don't see why it would be the case. Moreover, I suspect, as explained later, that better results are obtained without the regularisation because your model overfits.
>
> Reply: We are sorry for this confusion. Eq. 12 corresponds to the ‘non-informative prior $ p(\chi) \propto c $ as described in the paragraph above Eq. 12, while Eq. 11 corresponds to the prior in Eq. 8. The reviewer was correct that when applying Eq. 12, we are essentially relying on the implicit prior induced by the network - something that has been shown to be effective in prior work*. Based on our experiment, shown in Fig. 3, this implicit network prior works well for our problem.
> *Ulyanov et al. "Deep image prior." CVPR. 2018.
>
>
> 2. How do you estimate $\Sigma_{b'|\chi}$ in eq. 12?
>
> Reply: $\Sigma_{b'|\chi}$ was estimated when fitting the local field $b$ from multiple time delayed MR signal, voxelwise (Wang and Liu, 2015). Specifically, $b$ is the linear coefficient of such temporal fitting, and $\Sigma_{b'|\chi}$ is the fitting error of each voxel. So $\Sigma_{b'|\chi}$ is treated as a known parameter for the QSM dipole inversion problem
>
>
> 3. Experiments - it would seem that the unsupervised learning component via VI was trained and tested on the same data. If so, this is for me a major weakness on this work. The network may overfit on the testing data. Moreover, although you don't use any annotation for this task, this can be seen as an optimisation per subject. In this case, there is no clear advantage of using a deep learning approaches compared to other optimisation methods.
>
> Reply: The reviewer is correct that the unsupervised VI step in which training and testing was deployed on each hemorrhage patient case can be considered as a weakness, since its run-time is expensive. However, we note that this strategy only needs to be adopted  when we encounter a new type of case, e.g. a small number of hemorrhage cases unlike anything in the training data the model has seen before. Our point is that in such a scenario, one can use the unsupervised VI strategy (via Eq. 11 or 12 that do not rely on expensive COSMOS data) to further fine-tune/train the network. We expect that this fine-tuning needs to be done occasionally (on new types of cases - e.g. in a new patient population or after a scanner upgrade). After unsupervised fine-tuning, one can run a single forward pass for inference on future cases, which will be substantially more efficient than the MEDI benchmark. We have conducted some additional experiments to highlight this point.  Please see new Fig. 3 results here: http://gdurl.com/YTFy , where PDI-VI1 or PDI-VI2 refers to the above mentioned strategy training (fine-tuning)/validating on 5/1 hemorrhage cases and testing on 2 cases shown in new Fig. 3. Under-estimation issues inside hemorrhage in PDI are also reduced in PDI-VI1 and PDI-VI2 here. (We will also add a new benchmark FINE (Zhang et al., 2020) for comparison as requested by anonReviewer4). Please also see anonReviewer2’s detailed comments regarding the relevant questions and our response 1 for details.

---

### Meta-Review · Area_Chair1 · 2020-04-05
**MetaReview of Paper6 by AreaChair1**

**Rating:** 3
**Recommendation For Accepted Papers:** Poster

**Metareview:**

This paper proposes a Bayesian deep learning approach for solving Quantitative Susceptibility Mapping.

All reviewers agree that the paper is well written and the ideas and experiments are novel and interesting.

The validation is limited but enough in the opinion of the reviewers.



**Paper Type:**

methodological development

**Special Issue:**

no

---

### Decision · Program_Chairs · 2020-04-11

Accept